# A blueprint for robust crosslinking of mobile species in biogels with weakly adhesive molecular anchors

Jay Newby[1,2], Jennifer L. Schiller[3], Timothy Wessler[1,2], Jasmine Edelstein[4], M. Gregory Forest [1,2,4] & Samuel K. Lai[3,4,5]

Biopolymeric matrices can impede transport of nanoparticulates and pathogens by entropic or direct adhesive interactions, or by harnessing "third-party" molecular anchors to crosslink nanoparticulates to matrix constituents. The trapping potency of anchors is dictated by association rates and affinities to both nanoparticulates and matrix; the popular dogma is that long-lived, high-affinity bonds to both species facilitate optimal trapping. Here we present a contrasting paradigm combining experimental evidence (using IgG antibodies and Matrigel®), a theoretical framework (based on multiple timescale analysis), and computational modeling. Anchors that bind and unbind rapidly from matrix accumulate on nanoparticulates much more quickly than anchors that form high-affinity, long-lived bonds with matrix, leading to markedly greater trapping potency of multiple invading species without saturating matrix trapping capacity. Our results provide a blueprint for engineering molecular anchors with finely tuned affinities to effectively enhance the barrier properties of biogels against diverse nanoparticulate species.

[1] Department of Mathematics, University of North Carolina—Chapel Hill, Chapel Hill, NC 27599, USA. [2] Department of Applied Physical Sciences, University of North Carolina—Chapel Hill, Chapel Hill, NC 27599, USA. [3] Division of Pharmacoengineering and Molecular Pharmaceutics, Eshelman School of Pharmacy, University of North Carolina—Chapel Hill, Chapel Hill, NC 27599, USA. [4] UNC/NCSU Joint Department of Biomedical Engineering, University of North Carolina—Chapel Hill, Chapel Hill, NC 27599, USA. [5] Department of Microbiology & Immunology, University of North Carolina—Chapel Hill, Chapel Hill, NC 27599, USA. Jay Newby and Jennifer L. Schiller contributed equally to this work. Correspondence and requests for materials should be addressed to M.G.F. (email: forest@unc.edu) or to S.K.L. (email: lai@unc.edu)

Biopolymeric matrices are ubiquitous in living systems, generically composed of a highly entangled and crosslinked mesh of macromolecules in buffer. Within cells, cytoskeletal networks of actin and microtubules control cell migration, maintain cell shape and polarity, and facilitate proper routing and sorting of intracellular cargo[1, 2]. At the extracellular scale, networks of fibronectin, laminin, and collagen not only provide scaffolds for mechanical support and tissue organization but also regulate the dynamic behavior of cells through variations in local microstructure and stiffness[3, 4]. At the tissue scale, secreted mucins create a viscoelastic gel that serves both as a lubricant and as a transport barrier to prevent pathogens and particulates from reaching the underlying epithelium[5, 6].

A major function of biogels is to regulate transport. Gels can in theory impede the passive diffusion of particulates and viruses, as well as active motion of bacteria and cells, by steric obstruction and/or adhesive interactions to the matrix constituents[7]. Given that the majority of nanoscale species (henceforth referred to as nanoparticulates) are smaller than the mesh spacing of biogels, their diffusion across a gel barrier can only be hindered by adhesive interactions. However, due to evolutionary pressure, it is exceedingly unlikely that direct adhesive interactions with matrices comprised of relatively homogeneous constituents, such as mucins or laminins, can alone effectively block the transport of the full diversity of nanoparticulates typically encountered in nature. For example, viruses must penetrate the dense mucin mesh to infect underlying cells; thus, it is hardly surprising that the vast majority of viruses that transmit at mucosal surfaces (human immunodeficiency virus, herpes, human papillomavirus, Norwalk, etc.) are able to evade binding to mucins and diffuse rapidly through the low-viscosity interstitial fluids within pores of mucus gels[8].

An alternative strategy is to utilize "third-party" molecular anchors to crosslink nanoparticulates to the matrix, such as

antibodies (Abs) that can specifically recognize and bind invading pathogens. The diffusion coefficients of IgG and IgA Abs in human mucus are ~5–10% lower compared to buffer, whereas 10-fold larger viruses can diffuse in mucus unhindered[9]. The slightly retarded diffusion of both Abs implies they must be slowed by weak and transient interactions with the mucus matrix. Surprisingly, despite this seemingly negligible affinity, herpes-binding IgG can specifically and effectively immobilize herpes simplex virus type 1 (HSV-1) in human cervicovaginal mucus (CVM) even at sub-neutralizing IgG concentrations, and trapping HSV-1 in mucus directly prevented vaginal herpes transmission in mice[10]. Although the trapping potency of IgG is naturally affected by its binding and unbinding rates to mucins[11], the optimal kinetics remains poorly understood. The widely held and intuitively reasonable assumption is that anchors with long-lived, high-affinity bonds to both the nanoparticulate and matrix would confer superior trapping efficiency.

To develop more potent anchors, we seek to examine the characteristics of IgG that could maximize net adhesive interactions between nanoparticulates and biopolymer matrices. We combine theoretical models and experiments to show that anchor-matrix bonds that are rapid and short-lived relative to anchor-nanoparticulate bonds greatly enhance the trapping potency of molecular anchors.

## Results

**Efficient trapping with transient anchor-matrix bonds**. The highly viscoelastic nature of physiological mucus gels makes it exceedingly difficult to chemically modify and subsequently remove crosslinkers without irreversibly perturbing its rheological properties. Instead, we took advantage of the thermo-gelling properties of Matrigel®, which enables us to biotinylate the matrix as a low-viscosity fluid at 4 °C yet study its diffusional barrier

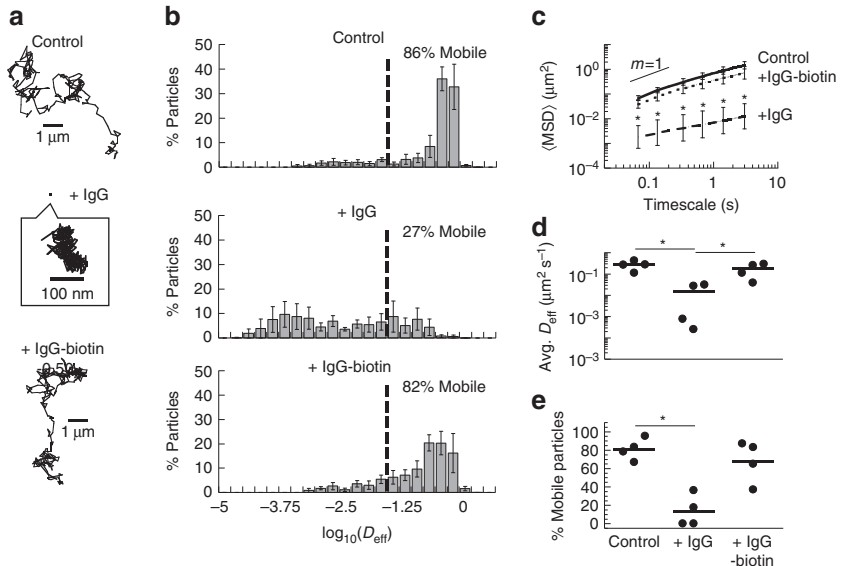

**Fig. 1** Transient vs. stable anchor-matrix bonds. Diffusion of 200 nm polyethylene glycol (PEG)-conjugated latex nanoparticles in biotinylated Matrigel® modified with neutravidin. **a** Representative traces of nanoparticles in Matrigel® with no added IgG (control), anti-PEG IgG (IgG), or biotinylated anti-PEG IgG (IgG-biotin) exhibiting effective diffusivities within one SEM of the ensemble average at a timescale of 1 s. **b** Distributions of the mean logarithms of individual particle effective diffusivities ($D_{eff}$) at a timescale of 0.2667 s. Log ($D_{eff}$) values to the left of the *dashed line* correspond to particles with displacements of less than 100 nm (i.e., roughly the particle diameter) within 0.2667 s. **c** Ensemble-averaged geometric mean square displacements (<MSD>) as a function of timescale, **d** mean $D_{eff}$ of all particles in each condition, and **e** fraction of mobile nanoparticles in Matrigel® treated with different IgG. $N = 4$ separately prepared slides/condition with 83–237 particles tracked per slide. *Error bars* represent SEM. *$p < 0.05$ compared to control in indicated comparisons. $p$ values were calculated by repeated measures two-way ANOVA in **c**, with one-way ANOVA on log-transformed data in **d**, and with one-way ANOVA in **e**

properties as a viscoelastic gel at 37 °C. IgG possess only modest affinity to Matrigel®, as reflected by its dissociation constant ($K_D \sim 4 \times 10^{-8}$ M; Supplementary Table 1) measured using biolayer interferometry (BLI). This allowed us to investigate, using anti-polyethylene glycol (PEG) IgG as molecular anchors, whether the mobility of PEG-modified polystyrene nanoparticles (PS-PEG; diameter ~200 nm) that exhibits rapid diffusion in the biotinylated Matrigel® can be altered by tuning the affinity of anchor-matrix bonds.

We mixed neutravidin and biotinylated IgG that specifically bind PEG into biotinylated Matrigel® to create high-affinity IgG-matrix bonds prior to temperature-induced gelation of the matrix; BLI measurements indicate biotinylated IgG exhibited much lower $K_D$ (~$1 \times 10^{-11}$ M; Supplementary Table 1) to biotinylated Matrigel® treated with neutravidin. When added to biotinylated Matrigel® mixed with neutravidin, either lacking exogenous IgG altogether or treated with control IgG, PS-PEG exhibited rapid diffusion, with a geometrically averaged ensemble effective diffusivity (<$D_{eff}$>; 0.25 $\mu m^2 s^{-1}$ at $\tau = 0.2667$ s) only ~3.2-fold reduced compared to their theoretical diffusivity in buffer (Fig. 1a, b). The conjugation of biotinylated IgG to Matrigel® did not reduce gel formation or the barrier properties of Matrigel® against ~200 nm uncoated carboxyl-modified nanoparticles, which were immobilized to a similar extent as in unmodified Matrigel® (Supplementary Fig. 1). Surprisingly, despite anchoring 10 $\mu g\,mL^{-1}$ anti-PEG IgG to Matrigel® with long-lived, high-affinity biotin-neutravidin bonds, the matrix largely failed to immobilize PS-PEG. Indeed, the <$D_{eff}$> (0.14 $\mu m^2 s^{-1}$) of PS-PEG at $\tau = 0.2667$ s was not statistically significantly different than in the same Matrigel® without anti-PEG IgG, and nearly 70% of particles remained mobile (defined as nanoparticles with <$D_{eff}$> $\geq 10^{-1.5}\ \mu m^2 s^{-1}$ at $\tau = 0.2667$ s; Fig. 1a–c). Modest trapping of PS-PEG by matrix-bound anti-PEG IgG was observed only with prolonged incubation, e.g., 24 h (Supplementary Fig. 2). These effects were not due to biotinylation of IgG or the presence of neutravidin with biotinylated IgG; anti-PEG IgG with and without biotinylation and/or neutravidin all exhibited similar $K_D$ (~$5 \times 10^{-9}$ M; Supplementary Table 2) as measured by BLI.

In contrast, despite the modest affinity between individual native unmodified IgG and biotinylated Matrigel®, the addition of 10 $\mu g\,mL^{-1}$ of anti-PEG IgG in biotinylated Matrigel® reduced the <$D_{eff}$> of PS-PEG by nearly 70-fold, with nanoparticles slowed on average 320-fold compared to their mobility in water (Fig. 1d). The fraction of mobile PS-PEG was reduced from 81 to 14% with the addition of anti-PEG IgG (Fig. 1e). The immobilization was not due to agglutination of PS-PEG, since trapped nanoparticles appeared identical to non-agglutinated nanoparticles in Matrigel® treated with control IgG. PS-PEG were also unlikely to be immobilized due to marked increase in the nanoparticle hydrodynamic diameter; a complete coating of IgG on ~200 nm nanoparticles would add no more than ~10 nm to the hydrodynamic diameter, and larger nanoparticles remained largely diffusive in Matrigel®. The presence of neutravidin was likewise not responsible for this phenomenon; native IgG trapped PS-PEG equally well in biotinylated Matrigel® whether treated with neutravidin or not (Supplementary Fig. 3). These results directly demonstrate that short-lived anchor-matrix bonds are far more efficient in facilitating immobilization of nanoparticles than long-lived anchor-matrix bonds. Finally, we found that addition of anti-PEG IgG also effectively immobilized smaller ~100 nm PS-PEG in Matrigel® (Supplementary Movies 1 and 2), and that addition of similar concentrations of Synagis®, a monoclonal IgG against protein F on respiratory syncytial virus (RSV), was able to effectively immobilize the virions in Matrigel® compared to control IgG (Supplementary Movies 3 and 4). These results

further illustrate the broad applicability of short-lived anchor-matrix bonds in facilitating trapping of diverse entities in biological matrices.

**Proposed theoretical framework and assumptions.** Our observations motivated us to develop a model to recapitulate the observations and examine the features of molecular anchors and matrix that could maximize trapping potency of nanoparticulates by the matrix. The model assumes three reactive species: molecular anchors A; nanoparticulates P; and matrix constituents M. Assuming that anchors must simultaneously possess some affinity to both the matrix and the nanoparticulates, our model reveals that the most robust crosslinking of nanoparticulates to the matrix, measured by a minimum effective particle diffusivity $D_{eff}$, is achieved when the following six conditions are met.

C1. Anchor-matrix binding-unbinding kinetics are markedly faster than anchor-nanoparticulate kinetics. In other words, the steady-state frequency of anchor-matrix binding ($\tau_{AM}^{-1}$) is high relative to the steady-state frequency of anchor-particle binding ($\tau_{AP}^{-1}$): $\tau_{AM}^{-1} \gg \tau_{AP}^{-1}$, or $\tau_{AM} \ll \tau_{AP}$.

C2. Nanoparticulates possess multiple ($N$) independent binding sites such that multiple anchors can simultaneously crosslink the same nanoparticulate to the matrix: $N \gg 1$.

C3. Anchors are much smaller than the nanoparticulate, and consequently, the anchor diffusivity ($D_A$) is much larger than the free nanoparticulate diffusivity ($D_P$): $D_A \gg D_P$.

C4. Anchor-nanoparticulate binding is fast enough (i.e., $\tau_{AP}$ is small enough) that many anchors are likely to bind to the nanoparticulate within the expected diffusive passage time ($\tau_L$) of nanoparticulate through the biopolymer matrix of thickness $L$: $\tau_{AP} < \tau_L$.

C5. For $[A] \gg [M]$, $\tau_{AM}$ is sufficiently short such that anchors do not saturate the binding sites in the matrix.

C6. Anchor concentration [A] is modest, such that on average a single nanoparticulate cannot be simultaneously bound by two anchors that are immobilized to the matrix. In other words, average [A] does not exceed one anchor per unit volume of the nanoparticulate ($V_P$): $[A] \ll 1/V_P$.

In general, with the proposed components M, A, and P, there are two reaction sequences that form the desired complex (MAP), corresponding to a trapped nanoparticulate. In particular, the MAP complex is formed either by a matrix-bound anchor capturing a free nanoparticulate:

$$M + A \underset{a_{off}}{\overset{a_{on}}{\rightleftarrows}} MA, \quad MA + P \underset{k_{off}}{\overset{k'_{on}}{\rightleftarrows}} MAP, \qquad (1)$$

or by a nanoparticulate-anchor complex (formed when free anchors accumulate on a diffusing nanoparticulate) interacting with and binding to the matrix:

$$A + P \underset{k_{off}}{\overset{k_{on}}{\rightleftarrows}} AP, \quad M + AP \underset{a_{off}}{\overset{\frac{D_P}{D_A}a_{on}}{\rightleftarrows}} MAP. \qquad (2)$$

The anchor-nanoparticulate binding rates for free anchors ($k_{on}$) and matrix-bound anchors ($k'_{on}$) are given by the Smoluchowski encounter relation[12], namely

$$k_{on} = (D_P + D_A)\varphi[A]R_0, \quad k'_{on} = (D_P + D_M)(1 - \varphi)[A]R_0, \qquad (3)$$

respectively, where $R_0$ is the effective binding distance at which two molecules react. Note that the diffusivity of the polymer matrix is effectively zero, i.e., $D_M \approx 0$. The fraction of free A at steady state is related to the binding ($a_{on}$) and unbinding ($a_{off}$)

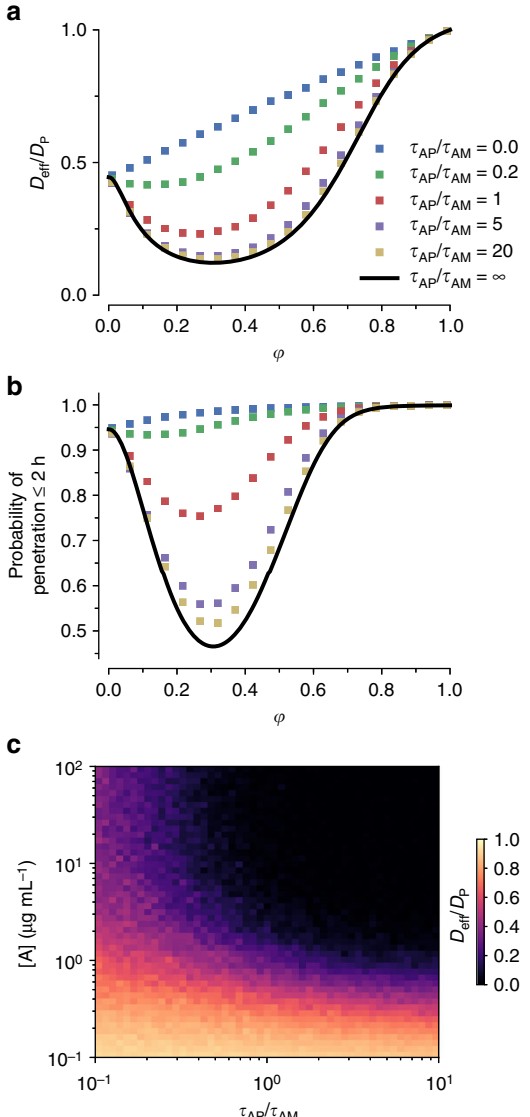

**Fig. 2** Monte-Carlo simulations of the effect of timescale separation on trapping. Timescale separation is represented by $\tau_{AP}/\tau_{AM}$. **a** Reduction in effective diffusivity of nanoparticulates as a function of the free fraction of anchors, $0 < \varphi < 1$, where 0 reflects anchors with permanent affinity and 1 reflects anchors with no affinity. **b** Probability of nanoparticulate penetrating across a layer of thickness $L = 50\ \mu m$ within 2 h. *Solid lines* show the $\tau_{AP}/\tau_{AM} \rightarrow \infty$ approximation. **c** Heat map of the effective diffusivity vs. the anchor concentration and timescale separation. Parameter values used: $D_A/D_P = 20$, $N = 15$

rates of anchors to the matrix, given by

$$\varphi = \frac{a_{off}}{a_{on} + a_{off}}.$$

Note that $\varphi = 0$ and $\varphi = 1$ represent extremes where all anchors and no anchors are bound to the matrix, respectively.

**How weak and rapid anchor-matrix binding maximizes trapping.** Instinctively, one may expect that $\varphi = 0$ maximizes the fraction of trapped nanoparticulates and that trapping potency is reduced as $\varphi$ rises until it is eliminated altogether when $\varphi = 1$. Nevertheless, this was not supported by our experiments where IgG, anchored to the matrix with long-lived biotin-avidin bonds, failed to trap nanoparticulates with the same potency as IgG that

exhibit only weak and short-lived interactions with the matrix. To begin to understand why long-lived anchor-matrix bonds may compromise nanoparticulate trapping, it is important to note that a nanoparticulate is unlikely to simultaneously encounter multiple matrix-bound (immobilized) anchors unless the anchor concentration is very high (i.e., anchors are generically spaced at distances much greater than the dimensions of the nanoparticulate). For example, we have previously observed trapping of ~100–200 nm nanoparticles and viruses at IgG concentrations of 1–3 µg mL$^{-1}$ in CVM[10, 13]; the average distance between each IgG at these concentrations is roughly 440–630 nm. At these concentrations, if anchors are permanently bound to the matrix, the average number of anchors on each 100–200 nm nanoparticle that has been crosslinked to the matrix must be at most one. Conversely, to achieve an average distance of ≤ 100 nm between each IgG would require IgG concentrations in excess of 250 µg mL$^{-1}$, an exceedingly high concentration for a single anchor species.

Recall from the Smoluchowski encounter relation that when anchors are immobilized, the rate of a nanoparticulate binding to an anchor is proportional to particle diffusivity $D_P$, whereas the binding rate of free anchors to the nanoparticulate is proportional to $D_P + D_A$. Since we postulated that $D_A \gg D_P$, nanoparticulates must encounter freely diffusing anchors much more frequently and quickly than matrix-bound anchors. Consequently, when $\varphi > 0$, multiple anchors begin to accumulate on the surface of the nanoparticulate, and multiple bonds can form (i.e., $PA_n \rightarrow P(AM)_n$) when a freely diffusing nanoparticulate-anchor complex encounters matrix constituents. While a single anchor might rapidly unbind from the matrix, resulting in a very short association lifetime of the complex, a nanoparticulate-anchor complex with multiple nanoparticulate-bound anchors, i.e., $PA_n$, can increase the collective crosslink lifetime because only one MAP bond is necessary to keep the nanoparticulate immobilized at any given time. So long as the anchors stay bound to the nanoparticulate, they do not diffuse away as quickly after the nanoparticulate-anchor complex unbinds from the matrix, as would individual free anchors, and thus can more rapidly rebind to the matrix. Assuming each anchor-matrix bond is independent, the $PA_n$ complex crosslink lifetime increases exponentially with the number of anchors $n$ bound to the same nanoparticulate, and it becomes exceedingly rare for all anchors to simultaneously unbind from the matrix. We therefore reach the seemingly counterintuitive conclusion that short-lived anchor-matrix bonds can actually facilitate more complete crosslinking of nanoparticulates to the matrix.

Of course, some fraction of A must bind to the polymer network with some probability or frequency; if $\varphi = 1$, then anchors never bind to the matrix. It follows that the crosslink lifetime of a nanoparticulate-anchor complex to the matrix must eventually begin to decrease as the anchor-matrix binding affinity is reduced below some optimal fraction of free anchors, $0 < \varphi < 1$, corresponding to the most robust crosslinking of nanoparticulates to matrix.

We seek to precisely identify the optimal affinity $\varphi$ and timescale of anchor-matrix interactions for minimizing nanoparticulate flux through a gel layer. To do so, we first define a characteristic length scale $L$ of interest in the system (e.g., the height of a mucus layer lining the surface of the lung or gastrointestinal (GI) tract). There are three timescales to consider: diffusion $(\tau_L = L^2/(2D_P))$; anchor-matrix interactions $(\tau_{AM} = 1/(a_{on} + a_{off}))$; and anchor-nanoparticulate interactions $(\tau_{AP} = 1/(D_A[A]R_0 + k_{off}))$. The diffusion timescale determines the average amount of time needed to diffuse through a matrix layer. The two kinetic timescales characterize the average duration of consecutive bind-unbind events. We assume that $L$

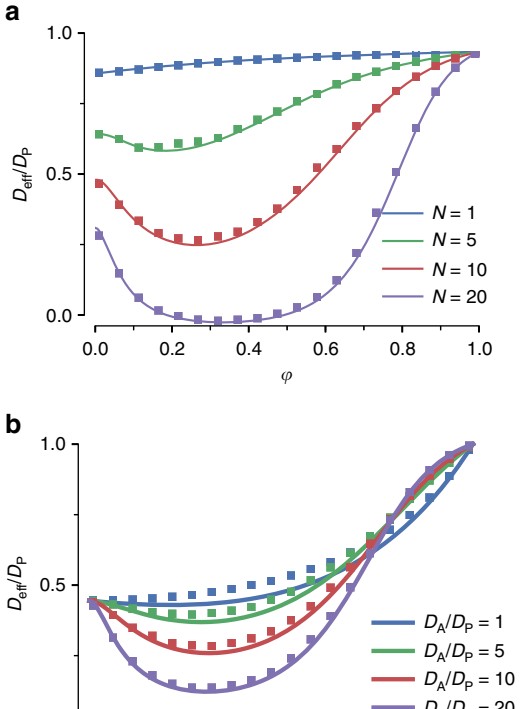

**Fig. 3** Effective diffusivity of nanoparticulates vs. free fraction of anchors. Monte-Carlo simulations varied values of **a** $N$, the maximum number of binding sites on the nanoparticle, and **b** $D_A/D_P$, the ratio of the anchor diffusivity to the nanoparticle diffusivity. Parameter values used were $\tau_{AP}/\tau_{AM} = 20$, $N = 15$, and $D_A/D_P = 20$

is large enough that many anchor-matrix and anchor-nanoparticulate interactions can occur before the nanoparticulates diffuse out of the system, i.e., $\tau_{AP}$, $\tau_{AM} \ll \tau_L$, which allows us to derive an effective diffusivity for the nanoparticulate $D_{eff} \leq D_P$ (see Methods section for the derivation) that characterizes the effective trapping potency of the anchors. The smaller the $D_{eff}$ is the more immobilized the nanoparticulate is: $D_{eff} = D_P$ indicates anchors that have no effect on the native diffusivity of the nanoparticulate, whereas $D_{eff} < D_P$ reflects anchors that can at least transiently immobilize the nanoparticulate to the matrix.

We next seek to explore how anchor-matrix affinity influences nanoparticulate trapping under two important regimes: slow but long-lived anchor-matrix kinetics, where $\tau_{AM} \gg \tau_{AP}$, and rapid yet short-lived anchor-matrix kinetics, where $\tau_{AM} \ll \tau_{AP}$. If $\tau_{AM} \gg \tau_{AP}$, then on the timescale of A/P kinetics, anchors do not bind to or unbind from the matrix. Because we assume that [A] is not unrealistically high (see above and assumption 6), nanoparticulates can effectively only bind to a single matrix-bound anchor at a time. In this regime, $D_{eff}$ is minimized when $\varphi = 0$, i.e., very high anchor-matrix affinity (Fig. 2a, $\tau_{AP}/\tau_{AM} \sim 0.01$). Assuming $D_A/D_P = 20$ and only 15 antigens on each 100 nm nanoparticulate, this results in a $D_{eff}/D_P$ that is reduced ~55% on average compared to if anchors have no affinity to matrix (Fig. 2a), which translates to only a ~5% reduction in the fraction of nanoparticulates that can penetrate a 50 µm-thick layer over 2 h (Fig. 2b).

A much different result is obtained with rapid and short-lived anchor-matrix bonds relative to anchor-nanoparticulate bonds, i.e., $\tau_{AM} \ll \tau_{AP}$. $D_{eff}/D_P$ for the same nanoparticulate drops significantly as $\tau_{AP}/\tau_{AM}$ increases, with a nontrivial optimal $\varphi$

that minimizes $D_{eff}$ (Fig. 2a). Indeed, when $\tau_{AP}/\tau_{AM}$ approaches 20 and with the steady-state free fraction of anchors in the ~20–40% range (i.e., $\varphi$ ~0.2–0.4), $D_{eff}/D_P$ is reduced by over 90%, effectively restricting transport of the nanoparticulates. This drop in $D_{eff}$ directly correlates to > 50% reduction in the fraction of nanoparticulates that can penetrate across a 50 µm-thick matrix layer over 2 h, a greater than 10-fold increase in trapping potency compared to the long-lived anchor-matrix bond scenario (Fig. 2b). Increases in $\tau_{AP}/\tau_{AM}$ also directly reduce the amount of anchors needed to suppress the flux of nanoparticulates penetrating and exiting the matrix layer (Fig. 2c). These results confirm condition C1 of our proposed model, namely that short-lived anchor-matrix bonds ($\tau_{AM} \ll \tau_{AP}$) maximize trapping potency.

**Other anchor features for maximizing trapping potency.** Hypothetically, if there is only one epitope available per nanoparticulate, then at most one anchor can bind to the nanoparticulate (i.e., $N = 1$). Naturally, in this scenario, $D_{eff}/D_P$ would decrease monotonically as $\varphi \rightarrow 0$, since maximum trapping is achieved when the nanoparticulate-bound anchor never dissociates from the matrix, as shown in Fig. 3a. In contrast, for $N > 1$, $D_{eff}/D_P$ is an exponentially decreasing function of $N$: the more antigen sites available on a nanoparticulate, the more likely and quickly the nanoparticulate will accumulate anchors on its surface and become trapped in the matrix (Fig. 3a). With even a modest number of anchor-binding sites on each nanoparticulate ($N$ ~20), nanoparticulate $D_{eff}$ can be reduced by over 90% when combined with rapid (i.e., $\tau_{AP}/\tau_{AM} = 20$) and weak (i.e., $\varphi$ ~0.2–0.4) anchor-matrix interactions. To place this in perspective, influenza and herpes simplex virus have hundreds of hemagglutinin[14] and gD glycoprotein[15] epitopes per viral particle, respectively. These results confirm condition C2 of our proposed model.

In addition to the number of binding sites, the rate of anchor accumulation depends on the frequency with which anchors can collide with the nanoparticulate. The latter is in turn proportional to the diffusivity of the anchor as predicted by the Smoluchowski encounter relation. Although greater nanoparticulate diffusivity $D_P$ can theoretically increase the encounter and anchor accumulation rate on the nanoparticulate, this also reduces the time $\tau_L$ available for sufficient quantities of anchor to accumulate on the nanoparticulate before the nanoparticulate diffuses through the barrier fluid. As shown in Fig. 3b, nanoparticulate $D_{eff}/D_P$ drops as $D_A$ increases; thus, anchors that are smaller and more mobile than the nanoparticulate are preferred for nanoparticulate trapping. These results confirm condition C3 of our proposed model.

**Effect of matrix thickness and anchor density on trapping.** The barrier properties of biogels are naturally dependent on both the thickness of the gel layer as well as the concentration of the molecular anchors. This is particularly relevant for diffusional barriers such as mucus and basement membranes, where minimizing the fraction of viruses that can penetrate through the gel layer can directly reduce the probability of transmission or spread of the infection systemically. To address the balance between these two parameters, we assert that the most effective balance of timescales to immobilize nanoparticulates is $\tau_{AM} \ll \tau_{AP} \ll \tau_L$ (i.e., imposing conditions C1 and C4 of our model). We have already shown above that maximal trapping occurs with rapid anchor-matrix binding kinetics i.e., $\tau_{AM} \ll \tau_{AP}$. In addition, as explained above, in order for nanoparticulates to become trapped in the matrix, the nanoparticulate must be captured by at least one anchor before it diffuses through the matrix. Recall that the average time the nanoparticulate,

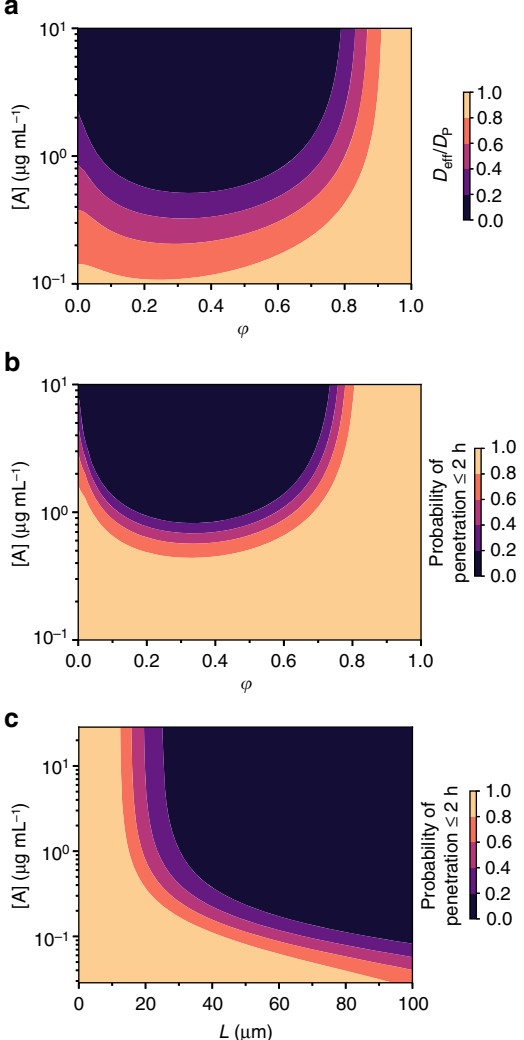

**Fig. 4** Maximizing trapping by varying matrix thickness and anchor concentration. Heat maps of the **a** effective diffusivity and **b**, **c** probability of penetration of a gel layer within 2 h as a function of different fraction of free anchors (**a**, **b**) and thickness of matrix layer (**c**). Parameter values used were as follows: **a** $D_A/D_P = 20$ and $N = 20$; **b** $D_A/D_P = 20$, $N = 20$, and $L = 50\,\mu m$; and **c** $D_A/D_P = 20$, $N = 20$, and $\varphi = 0.7$

(typically ~50–100+ μm), as well as (ii) the Ab concentrations present in mucus (typically ~0.1–10 μg mL$^{-1}$)[10]. This analysis confirms condition C4 of our proposed model.

**Robust trapping of multiple nanoparticulate species**. To selectively control transport against multiple species of nanoparticulates in the same polymeric matrix, such as trapping a diverse array of pathogens that impinge on mucus coating the airways and GI epithelium, many corresponding anchor species must coexist without impeding each other's trapping potency. In other words, even when anchors that bind any given species represent only a tiny fraction of all anchors present, the specific anchor-matrix affinity must remain unaltered in order to maintain comparable trapping potency. We introduce the term "trapping robustness" to describe the ability to immobilize multiple nanoparticulate species.

Since the concentration of matrix constituents is finite, the number of anchor-binding sites in a polymeric gel must by definition be finite. Thus, if anchor-matrix bonds are long-lived, a matrix-bound anchor prevents other anchors from binding to the same binding site on the matrix. Thus, at concentrations sufficient for trapping (e.g., ~1–5 μg mL$^{-1}$ IgG), the system could accommodate only a relatively limited number of anchor species ($< 10^3$ for a 2% w/v gel, assuming 10 anchor-binding sites per matrix molecule and an average molecular weight of 500 kDa) before additional anchors become unable to effectively reduce $D_{eff}/D_P$ (Fig. 5a). However, when $\tau_{AM} \ll \tau_{AP}$, the short duration of anchor-matrix bonds would greatly increase the number of unoccupied anchor-binding sites available on the matrix at any moment in time. This in turn enables the anchor-matrix system to both immobilize a far greater number of particle species simultaneously as well as reduce the minimum $D_{eff}/D_P$ that could be achieved with each anchor (Fig. 5b, c). Indeed, when $\tau_{AP}/\tau_{AM} \geq 20$, a biogel reinforced with appropriate molecular anchors can effectively immobilize at least 30-fold more (i.e., ~$3 \times 10^4$) distinct nanoparticulate species without appreciable loss in trapping potency (i.e., similar minimum $D_{eff}/D_P$ for all species), underscoring the potential trapping robustness of the system. Altogether, these results confirm condition C5 of our proposed model.

Finally, we examined the relative importance for each of the parameters described above by evaluating the partial derivatives of $\log(D_{eff}/D_P)$ under a range of parameter values. We observed the greatest impact with variations with $\tau_{AP}/\tau_{AM}$ (in particular at low $\tau_{AP}/\tau_{AM}$ values) and with the rates of anchor accumulation $k_{on}[A]$, with more modest impact with changes in antigenic epitope density (i.e., maximum number of bound anchor) $N$ and the diffusivity of anchors relative to the nanoparticulate species $D_A/D_P$ (Fig. 6). These results underscore short-lived anchor-matrix bonds relative to anchor-nanoparticulate bonds as a crucial separation of timescales for enabling molecular anchors that can substantially enhance the barrier properties of either biological or synthetic polymeric matrices to multiple nanoparticulate species.

**Discussion**. A critical function of polymeric matrices in biological systems is to exert selective control over the transport of thousands of nanoparticulate species. By eliminating the need for matrix constituents to directly recognize diverse antigenic species, an anchor-matrix system can enable an effective diffusional barrier against many nanoparticulate species while maintaining relatively static biochemistry and microstructure of the matrix. This suggests that anchors, such as IgG and other Abs produced by the immune system that can adapt and bind diverse molecular entities, represent an ideal platform to control nanoparticulate

unhindered by anchors, needs to diffuse through a matrix layer is $\tau_L = L^2/(2D_P)$. Hence, the matrix layer thickness must be $L \gg \sqrt{2D_P\tau_P}$ for anchors to have sufficient time to accumulate on the nanoparticulate. To illustrate the effect of the timescale $\tau_D$, we use a numerical approximation of the solution to Eq. (15) (see Methods section), and compute the probability that a nanoparticulate can diffuse across a polymer matrix layer of thickness $L$. We term this the absorption probability; a low absorption probability indicates effective trapping by anchors. Not surprisingly, $D_{eff}/D_P$ (Fig. 4a) and the absorption probability (Fig. 4b) both decrease with increasing anchor concentration (Fig. 4a), in both cases approaching a minimum $D_{eff}/D_P$ when $\varphi$ ~0.35. Interestingly, when we compare the relative importance of $L$ vs. anchor concentrations, we found that exponentially higher anchor concentrations are required when $L$ is smaller than ~40–50 μm thick in order to maintain a comparably effective diffusional barrier (Fig. 4c), implying that effective diffusional barriers in vivo should be at least 40–50 μm thick. These estimates agree remarkably well with both (i) the thickness of mucus coatings lining the respiratory, GI, and cervicovaginal tracts

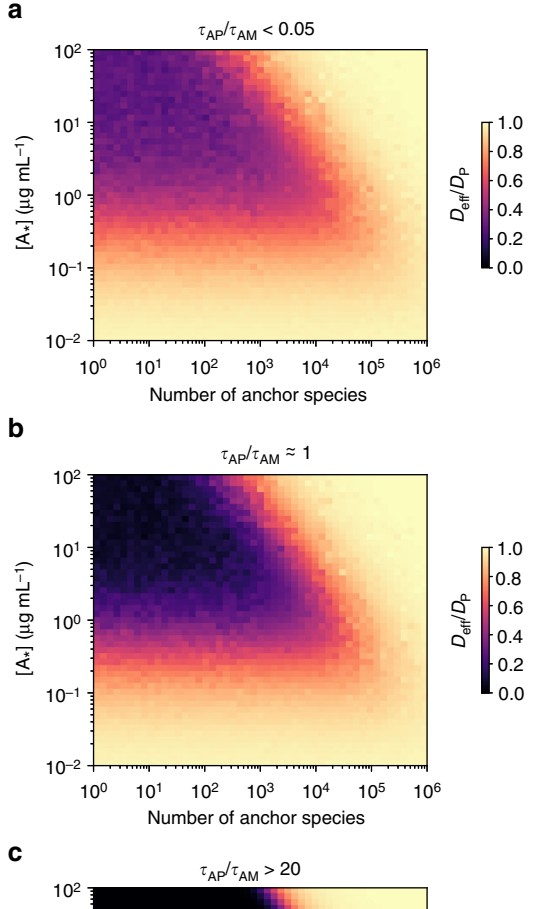

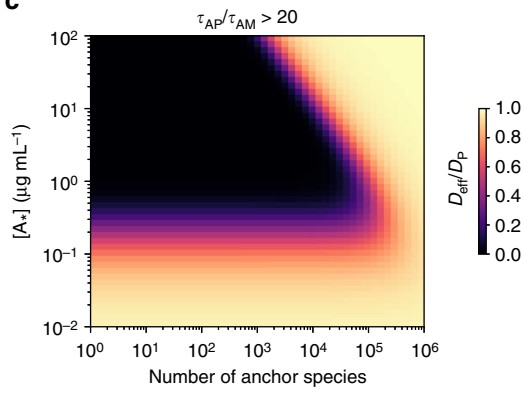

**Fig. 5** Potential saturation of trapping potency of molecular anchors. Potential saturation of trapping potency was measured by effective diffusivity of nanoparticulates, in the context of molecular anchors in matrix containing other anchor species: **a** $\tau_{AP}/\tau_{AM} = 0.05$; **b** $\tau_{AP}/\tau_{AM} = 1$; and **c** $\tau_{AP}/\tau_{AM} = 20$. Parameter values used were $D_A/D_P = 20$, $N = 20$, and $[M] = 10^5\ \mu m^{-3}$ (which corresponds to a 2% w/v gel with 10 anchor-binding sites per matrix)

transport. Here we demonstrate both experimentally and theoretically that short-lived anchor-matrix interactions convey the key attribute enabling potent and robust control over nanoparticulate transport in biogels (schematic in Fig. 7). Combined with our earlier observations that IgG can potently immobilize viruses and nanoparticles in different mucus secretions, it is likely that the proposed strategy, whereby the barrier properties are tuned by modest concentrations of highly mobile molecular anchors with exceedingly short anchor-matrix bond times relative to anchor-nanoparticulate bond times, is a universal feature of biogels in living systems. Our findings provide, for the first time, a blueprint for engineering of

molecular anchors with optimal short-lived anchor-matrix and anchor-nanoparticulate bonds to selectively and potently tune the barrier properties of polymeric gels. These insights will help guide the development of methods to reinforce the natural biological barriers against pathogens, such as the mucus barrier against sexually or respiratory transmitted infections.

## Methods

**Preparation of PEG-coated nanoparticles.** To produce PEGylated nanoparticles (PS-PEG), we covalently modified 200 nm fluorescent, carboxyl-modified polystyrene beads (PS-COOH; Invitrogen) with 2 kDa methoxy poly(ethylene glycol) amine (PEG; Sigma) via a carboxyl-amine reaction, as published previously[16, 17]. Particle size and $\zeta$-potential were determined by dynamic light scattering and laser Doppler anemometry, respectively, using a Zetasizer Nano ZS (Malvern Instruments, Southborough, MA, USA). Size measurements were performed at 25 °C at a scattering angle of 90°. Samples were diluted in 10 mM NaCl solution, and measurements were performed according to the instrument instructions. High-density PEGylation (> 1 PEG per nm²) was verified using the fluorogenic compound 1-pyrenyldiazomethane to quantify residual unmodified carboxyl groups on the polystyrene beads[17]. PEG conjugation was also confirmed by a near-neutral $\zeta$-potential (Supplementary Table 1)[16].

**Preparation of biotinylated Matrigel®.** Growth factor-reduced Matrigel® (Corning) was dialyzed against phosphate-buffered saline (PBS) for a minimum of 24 h at 4 °C, then biotinylated with 20-fold molar concentration NHS-PEG4-biotin (Thermo Fisher), which was again dialyzed against PBS for a minimum of 24 h at 4 °C. This biotinylated Matrigel® (final concentration 2.2 mg mL⁻¹) was mixed with neutravidin (Thermo Fisher; final concentration 0 or 4 µg mL⁻¹), bovine serum albumin (Sigma, final concentration 1 mg mL⁻¹), and Eagle's minimum essential medium (Lonza BioWhittacker), on ice for 15 min. Fluorescent PS-COOH (Ex: 505 nm, Em: 515 nm; final concentration $4.5 \times 10^8$ beads per mL) and PS-PEG (Ex: 625 nm, Em: 645 nm; final concentration $4.3 \times 10^8$ beads per mL) nanoparticles and anti-PEG IgG₁ (CH2076 or CH2076B, Silver Lake Research, final concentration 10 µg mL⁻¹) was combined on ice. The mixture was added to a custom-made micro-volume glass chamber slide, incubated at 37 °C for 45 min in a custom hydration chamber, then sealed and incubated for another 30 min prior to microscopy. Trapping of PS-COOH beads in Matrigel® was used as an internal control in all microscopy experiments as a measure of complete polymerization of Matrigel® constituents.

**Preparation of fluorescently labeled RSV.** RSV was fluorescently labeled with AlexaFluor 555 via a $N$-hydroxysuccinimide ester reaction. Briefly, 200 µL of RSV (2 mg mL⁻¹) were diluted in 20 µL of 1 M bicarbonate buffer, then added to 20 µL of AlexaFluor 555 NHS Ester (1 mg mL⁻¹). The reaction was incubated in the dark with gentle rocking for 2 h, dialyzed (molecular weight cutoff 100 kDa) against $1 \times$ PBS at 4 °C to remove unbound fluorophores, and stored at − 80 °C[18]. The Pierce BCA Protein Assay Kit (Thermo Scientific) was used to assess RSV concentration post labeling.

To verify that labeling did not significantly affect Ab binding, an enzyme-linked immunosorbent assay was run with IgG that bind to different RSV epitopes. High-affinity half-area 96-well Costar plates (Corning) were coated with 50 µL of unlabeled or labeled RSV (10 µg mL⁻¹), then incubated overnight at 4 °C. After blocking plates with 5% non-fat milk in PBS, 50 µL of anti-RSV IgG (Synagis (MedImmune NDC 60574-4114-1), MAB8599 (EMD Millipore ca. no. MAB8599), and MAB8582 (EMD Millipore ca. no. MAB8582)) at various concentrations in 1% milk were added to the corresponding wells. Antibodies bound to virus were detected with a horseradish peroxidase (HRP)-conjugated IgG diluted 1:10,000 in 1% milk (F(ab')2 anti-mouse IgG Fc (Goat)-HRP conjugate (Santa Cruz Biotechnology cat. no. SE-2005) and F(ab')2 anti-human IgG Fc (Goat)-HRP conjugate (Rockland cat. no. 7091317)). A volume of 50 µL 1-Step Ultra TMB (ThermoFisher) was the HRP substrate, then the reaction was quenched with 50 µL of 2 N sulfuric acid. Absorbance at 450 nm was measured using a Spectramax M2 plate reader (Molecular Devices). All wash and incubation steps were performed using PBS with 0.05% Tween.

**High-resolution multiple particle tracking.** The trajectories of the fluorescent particles were recorded using an electron-multiplying charge-coupled device (EMCCD) camera (Evolve 512; Photometrics, Tucson, AZ) mounted on an inverted epifluorescence microscope (AxioObserver D1; Zeiss, Thornwood, NY, USA), equipped with an Alpha Plan-Apo ×100/1.46 numerical aperture objective, environmental (temperature and $CO_2$) control chamber, and an LED light source (Lumencor Light Engine DAPI/GFP/543/623/690). Twenty-second videos (512 × 512, 16-bit image depth) were captured with MetaMorph imaging software (Molecular Devices, Sunnyvale, CA, USA) at a temporal resolution of 66.7 ms and spatial resolution of 10 nm (nominal pixel resolution 0.156 µm pixel⁻¹). The tracking resolution was determined by tracking the displacements of particles immobilized with a strong adhesive, following a previously described method[19]. Particle trajectories were analyzed using MATLAB software as described

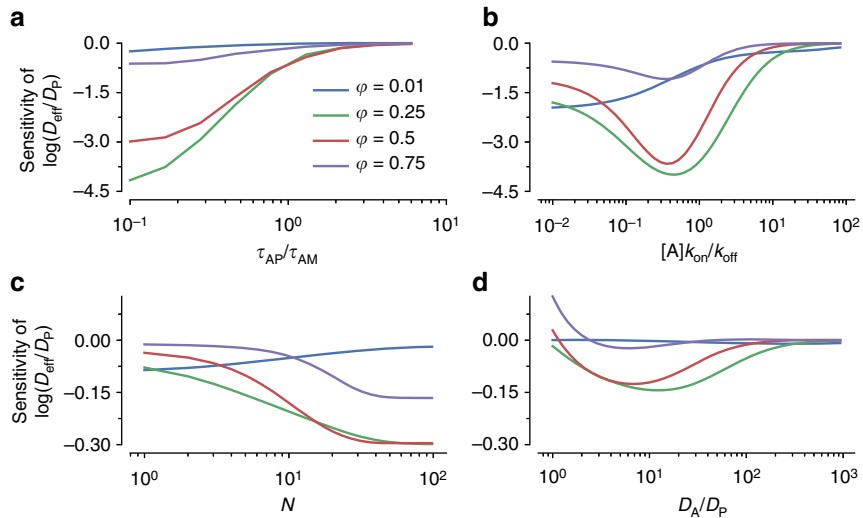

**Fig. 6** Sensitivity analysis for parameters that influence the trapping of anchors. Parameters analyzed were **a** $\tau_{AP}/\tau_{AM}$, **b** $[A]k_{on}/k_{off}$, **c** $N$, and **d** $D_A/D_P$. Sensitivity is defined as $\frac{\partial}{\partial p}\log\left(\frac{D_{eff}}{D_P}\right) \approx \frac{\Delta D_{eff}}{D_{eff}\Delta p'}$, where $p$ can be $\tau_{AP}/\tau_{AM}$, $[A]k_{on}/k_{off}$, $N$, or $D_A/D_P$. Fixed parameters were $D_A/D_P = 20$, $N = 20$, $\tau_{AP}/\tau_{AM} = 20$, and $[A]k_{on}/k_{off} = 2$

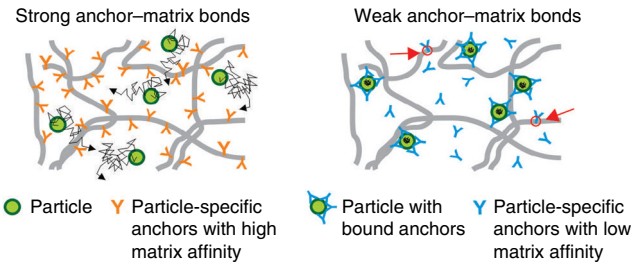

Strong anchor–matrix bonds          Weak anchor–matrix bonds

○ Particle   Y Particle-specific anchors with high matrix affinity

● Particle with bound anchors   Y Particle-specific anchors with low matrix affinity

**Fig. 7** Schematic of proposed mechanism. Anchors with weak and rapid interactions with the matrix can facilitate more effective crosslinking of nanoparticulates to matrix constituents compared to anchors with high-affinity, long-lived interactions with the matrix. The extent that the motion of nanoparticulates are hindered by different anchors are reflected by the dimensions of their traces (*black*). *Red circles/arrows* indicate the small fraction of free (not particle-bound) anchors that will transiently interact with the matrix at any moment in time

previously[20]. Sub-pixel tracking resolution was achieved by determining the precise location of the particle centroid by light-intensity-weighted averaging of neighboring pixels. Trajectories of $n \geq 40$ particles per frame on average (corresponding to $n \geq 80$ total traces per specimen per condition) were analyzed for each experiment, and 3–4 independent experiments were performed for each condition. Under the assumption that particle paths are samples of a stationary stochastic process, mean squared displacement (MSD) can be computed by time averaging, i.e., $\langle \Delta r^2(n\Delta t)\rangle = \frac{1}{N-n}\sum_{m=1}^{N-n}\{[x((m+n)\Delta t) - x(m\Delta t)]^2 +$ $[y((m+n)\Delta t) - y(m\Delta t)]^2\}$, (where $\tau$ = timescale or time lag $\Delta t$ is the time between video frames), and $\tau = n\Delta t$). MSDs from all particle paths within a given video were then ensemble averaged to obtain $\langle MSD\rangle$. Distributions of MSDs and effective diffusivities ($D_{eff}$) were calculated as previously demonstrated[16]. MSD may also be expressed as MSD $= 4D_0\tau^\alpha$, where $\alpha$, the slope of the curve on a log-log scale, is a measure of the extent of impediment to particle diffusion ($\alpha = 1$ for pure unobstructed Brownian diffusion; $\alpha < 1$ indicates sub-diffusive motion due to interactions with the elastic as well as viscous properties of the polymeric gel). Mobile particles were defined as those with $D_{eff} \geq 10^{-1.5}$ μm$^2$ s$^{-1}$ at $\tau = 0.2667$ s (this $\tau$ corresponds to a minimum trajectory length of five frames), based on multiple data sets of mobile and immobile nanoparticles (e.g., PS and PS-PEG nanoparticles) in Matrigel®[16, 21].

**BLI experiments.** Similar to published protocols[22], on an Octet QK instrument (ForteBio), streptavidin biosensors (ForteBio) were loaded with ligand and blocked with free biotin. To measure Ab affinity with Matrigel®, sensors were loaded with biotinylated Matrigel®, and to measure Ab affinity with antigen, 10 kDa biotin-PEG-NH$_2$ was loaded onto sensors. Abs (biotinylated and native) at different

concentrations were associated with these customized biosensors and dissociated into running buffer. Data were adjusted for reference sensors and baseline values and aligned to dissociation, then processed with Savitzky–Golay filtering. Analysis was performed with ForteBio software using a 1:1 global curve fit model to obtain values for $k_{on}$, $k_{off}$, and $K_D$.

**Mathematical model.** Instead of developing a reaction-diffusion model for a concentration of nanoparticulate species P, we take a stochastic approach and focus on the motion of individual nanoparticle P. The goal is to maximize the fraction of time that P spends bound to the polymer network.

Let $N$ be the total number of anchor-binding sites on the nanoparticle P. The crosslink enhancement effect requires the cooperative action of multiple anchors; it requires $N \gg 1$ anchor-binding sites (e.g., antigenic epitopes) on the nanoparticle. The reaction Eqs. (1) and (2) describe the case of a single binding site (i.e., $N = 1$). The first crosslink bond forms at a diffusion-limited reaction rate according to Eq. (2). If $N > 1$, additional anchors might be bound to the nanoparticle. Once the first crosslink forms many additional binding sites M are very close by, allowing additional anchor-matrix bonds to form. The intra-complex reaction is given by

$$\text{M} + \text{AP(AM)}_{n-1} \underset{na_{off}}{\overset{Ca_{on}}{\rightleftarrows}} \text{P(AM)}_n, \quad n > 1, \tag{4}$$

where $C$ is a nondimensional parameter that scales the intra-complex binding rate and assumed to be 1 in the current work. Molecules within a large complex may not react with each other at the same rate as they do when they are freely diffusing. When nanoparticulates, anchors, and matrix elements are bound within the same complex, they are mechanically linked. Mechanical forces imposed by surrounding elements of the complex confine random molecular motion[23]. Similar biomechanical reactions are common in biology (e.g., molecular motor transport[24] and DNA transcription[25, 26]). For our present situation, it is a reasonable first approximation to assume that the intra-complex dissociation rates remain the same as the bimolecular dissociation rates (i.e., $a_{off}$ and $k_{off}$). However, intra-complex binding rates are different from the Smoluchowski bimolecular reaction rates for diffusing molecules. The binding rate between two molecules within the complex depends on their relative distance and effective random mobility. Molecules within a single nanocomplex are quite close so that they do not have to move far in order to bind. On the other hand, they have lower relative mobility when mechanically confined within the complex.

Let $n$ be the number of occupied binding sites, and let $s$ be the number of anchors crosslinking the nanoparticulate to the polymer network. The chemical system can be modeled as a Markov process with state transitions given by

$$n \underset{(n-s+1)k_{off}}{\overset{(N-n)k_{on}}{\rightleftarrows}} n+1, \quad (s, n) \underset{(s+1)k_{off}}{\overset{\delta_{s,0}(N-n)k'_{on}}{\rightleftarrows}} (s+1, n+1), \tag{5}$$

$$s \underset{(s+1)a_{off}}{\overset{g(s)(n-s)a_{on}}{\rightleftarrows}} s+1, \tag{6}$$

where

$$g(s) = \begin{cases} \frac{D_P}{D_A}, & s = 0 \\ C, & s \geq 1 \end{cases}. \tag{7}$$

The process is described by its probability density $p(n, s, x, t)$. The probability that at time $t$, the nanoparticulate is bound to $n$ anchors, $s$ of which are bound to elements of the polymer matrix, and located within a small distance $dx$ of position $x$ is

$$\mathrm{Prob}[n(t) = n, s(t) = s, x < x(t) < x + dx] \approx p(n, s, x, t)dx + o(dx). \tag{8}$$

Since the model is a continuous time Markov process, the probability density function satisfies the differential Chapman–Kolmogorov equation (see ref. [27] for details of the derivation),

$$\frac{\partial}{\partial t} p(n, s, x, t) = \delta_{s,0} D_P \nabla^2 p + [\mathbb{M}_s + \mathbb{V}_{n,s}] p, \tag{9}$$

where $\mathbb{M}_s$ is the $N \times N$ transition rate matrix for Eq. (6) and $\mathbb{V}_{n,s}$ is the $N \times N$ transition rate matrix for Eq. (5).

**Multiple timescale analysis.** Consider the case where $\tau_M \ll \tau_P$. Notice that the fast reaction Eq. (6) conserves $n$. While $n$ changes slowly, transitions in $s$ are at quasi steady state. Since we are primarily concerned with the motion of the nanoparticulate, and not necessarily the state of any bound anchors, our goal is to obtain a good approximation to the marginal probability

$$u(x, t) = \sum_{n=0}^{N} \sum_{s=0}^{N} p(n, s, x, t). \tag{10}$$

Using the rule of conditional probability, we can rewrite the full probability density for the process as

$$p(n, s, x, t) = \rho_t(s, t | n, x) \rho_t(n, t | x) u(x, t). \tag{11}$$

Since $s$ changes rapidly compared to $n$, which changes rapidly compared to $x$, the conditional probabilities $\rho_t$ rapidly equilibrate, which means that $\rho_t \approx \rho_\infty = \rho$. Since the transition rates are independent of position $x$, it follows that $\rho(s | n, x) = \rho(s | n)$ and $\rho(n | x) = \rho(n)$. The two probability distributions $\rho(s | n)$ and $\rho(n)$ are called quasi-steady-state distributions, and they satisfy

$$\sum_{s=0}^{N} \mathbb{M}_s \, \rho(s | n) = 0, \tag{12}$$

$$\sum_{s=0}^{N} \sum_{n=0}^{N} \mathbb{V}_{n,s} \, \rho(s | n) \rho(n) = 0. \tag{13}$$

We can take advantage of the separation of timescales with an asymptotic approximation, namely

$$p(n, s, x, t) \sim \rho(s | n) \rho(n) u(x, t). \tag{14}$$

First we average out the fastest reaction, the transition in $s$. Let $\bar{p}(n, x, t) = \rho(n) u(x, t)$. Substituting Eq. (14) into Eq. (9), summing over $s$, and using Eq. (13) yields

$$\frac{\partial}{\partial t} \bar{p}(n, x, t) = D_P \rho(0 | n) \nabla^2 \bar{p} + \overline{\mathbb{V}}_n \bar{p}, \tag{15}$$

where $\overline{\mathbb{V}}_n$ is the transition rate matrix for the averaged slow reaction:

$$n \underset{(n+1)k_{\mathrm{off}}}{\overset{(N-n)\kappa(n)}{\rightleftharpoons}} n + 1, \tag{16}$$

where $\kappa(n) = \rho(0|n)k_{\mathrm{on}} + k'_{\mathrm{on}}$. Given $n$, the stationary distribution for the number of anchors $s < n$ on the nanoparticulate that are bound to the matrix is

$$\rho(s|n) = \begin{cases} \frac{D_A C \alpha^n}{(D_A C - D_P)\alpha^n + D_P}, & s = 0 \\ \frac{D_P \binom{n}{s}(1-\alpha)^s \alpha^{n-s}}{(D_A C - D_P)\alpha^n + D_P}, & s > 0, \end{cases} \tag{17}$$

where $\alpha = a_{\mathrm{off}}/(Ca_{\mathrm{on}} + a_{\mathrm{off}})$.

We can apply the same procedure to average out $n$ as follows. The quasi-steady-state distribution for $n$ is given by

$$\rho(n) = \frac{\mathcal{N}}{k_{\mathrm{off}}^n} \binom{N}{n} \prod_{j=0}^{n-1} \left( \rho(0|j)k_{\mathrm{on}} + k'_{\mathrm{on}} \right), \tag{18}$$

where $\mathcal{N}$ is a normalization factor. Substituting $\bar{p}(n, x, t) = \rho(n) u(x, t)$ into Eq. (15), summing over $n$, and using Eq. (13) yields

$$\frac{\partial}{\partial t} u(x, t) = D_{\mathrm{eff}} \nabla^2 u, \qquad D_{\mathrm{eff}} = D_P \sum_{n=0}^{N} \rho(0|n)\rho(n). \tag{19}$$

**Monte-Carlo simulations.** To determine the accuracy of the above approximation, we use Monte-Carlo simulations. Using the Gillespie algorithm[28], we simulate the Markov chain Eqs. (4) and (5) (which is independent of $x$). A single realization is generated through $m$ state transitions. The total elapsed time $t_m$ and the total time spent with $s = 0$ (the free diffusing state) $t_m^{(0)}$ are updated with each transition. It is easy to show that

$$P[s = 0] = \lim_{m \to \infty} \frac{t_m^{(0)}}{t_m}. \tag{20}$$

Because all increments from free diffusion are independent, an estimator for the effective diffusivity is

$$D_m^{\mathrm{eff}} \equiv D_P \frac{t_m^{(0)}}{t_m}. \tag{21}$$

**Saturated regime.** The reaction rate for any individual free anchor is substantially reduced by the saturation of matrix-binding sites. Because $[A_T] \gg [M]$, the fraction of unoccupied matrix-binding sites $\xi$ is equivalent to the fraction of time an individual matrix-binding site is unoccupied. Hence,

$$\xi = \frac{a_{\mathrm{off}}}{\frac{[A_T]}{[M]} a_{\mathrm{on}} + a_{\mathrm{off}}} \approx \frac{a_{\mathrm{off}}[M]}{a_{\mathrm{on}}[A_T]}. \tag{22}$$

It follows that the binding rate for an individual freely diffusing anchor is

$$a'_{\mathrm{on}} = \xi a_{\mathrm{on}} \approx \frac{a_{\mathrm{off}}[M]}{[A_T]}. \tag{23}$$

Similarly, the intra-complex binding rate (see Eq. (3)) is $a''_{\mathrm{on}} = C a_{\mathrm{off}}[M]/[A_T]$. Based on the modified binding rate (Eq. (11)), the anchor-matrix kinetic timescale becomes

$$\tau_{\mathrm{AM}} = \frac{1}{\left(1 + \frac{[M]}{[A_T]}\right) a_{\mathrm{off}}} \approx 1/a_{\mathrm{off}}. \tag{24}$$

The effective diffusivity in the saturated regime is obtained by substituting $[A] = [A_*]$, $\varphi \approx 1 - \frac{[M]}{[A_T]}$, and $\alpha \approx 1 - C\frac{[M]}{[A_T]}$ into Eq. (19).

**Statistics.** MSD data were log-transformed and compared within groups using a repeated-measures two-way analysis of variance (ANOVA) and post hoc Šidák test. Log-transformed average $D_{\mathrm{eff}}$ and non-transformed % mobile were compared with one-way ANOVA and subsequent Tukey's honest significant difference tests. In all analyses, global $\alpha = 0.05$. Error bars and $\pm$ represent SEM.

**Code availability.** Modeling results and simulations were performed using Python and C. Plotting was done using Python libraries: Jupyter, Scipy, and Matplotlib. Monte-Carlo simulations were done in C using the GNU Scientific Library for random number generators. Particle trajectories were analyzed using a MATLAB version of open-source particle-tracking code, originally developed in IDL by Crocker and Hoffman[29]. We have adapted this code to analyze particle trajectories on a "frame-by-frame" basis, as described previously[20].

**Data availability.** The data sets generated during and/or analyzed during the current study are available from the corresponding author on reasonable request.

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

## Acknowledgements

This work was supported by Eshelman Institute for Innovation, National Institutes of Health (http://www.nih.gov/) Grants R21EB017938 (S.K.L.); the National Science Foundation (http://www.nsf.gov/) DMS-1412844 (M.G.F.), DMS-1462992 (M.G.F.), and CAREER Award DMR-1151477 (S.K.L.); the David and Lucile Packard Foundation (http://www.packard.org/) (2013-39274; S.K.L.); startup funds from the Eshelman School of Pharmacy (https://pharmacy.unc.edu/) and Lineberger Comprehensive Cancer Center at the University of North Carolina—Chapel Hill (https://unclineberger.org/) (S.K.L.); and PhRMA Foundation Predoctoral Fellowship (J.L.S.). The funders had no role in study design, data collection and analysis, decision to publish, or preparation of the manuscript. Finally, we thank Richard Cone for insightful discussions.

## Author contributions

J.N., M.G.F., and S.K.L. conceived of and developed the theoretical framework; J.L.S. and S.K.L. designed the experiments and analyzed the data; J.L.S. performed the experiments; J.E. prepared and verified the RSV; J.N. and T.W. performed computational modeling/simulations; J.N., J.L.S., M.G.F., and S.K.L. wrote the paper.

## Additional information

**Competing interests:** S.K.L. is the founder of, serves on the board of directors for, and maintains a financial interest in Mucommune, which is actively seeking to commercialize "muco-trapping" antibody technology. The terms of these arrangements are being managed by The University of North Carolina in accordance with its conflict of interest policies. The remaining authors declare no competing financial interests.

