## [Peer Review File · Nature Communications]

Reviewers' comments:

Reviewer #1 (Remarks to the Author):

Newby and others examined the characteristics of IgG (a representative third party molecular anchor) that could potentially maximize the net adhesive interactions between nanoparticulates and biopolymer hydrogel matrix. They found a mechanism, different from previously widely held assumption, that only rapid and short-lived anchor-nanoparticulate bonds (not the long-lived and high-affinity bonds) will greatly enhance the trapping potency of the molecular anchors. They applied a mathematic modeling to propose the unique mechanism that has been evaluated through experimental data. Their findings are novel and will be interesting to the scientific community specifically in mucus biology and mucosa-related pathophysiology. The findings form the foundation for better design and development of more potent anchors that can enhance the pathogen tracking in the biological gel system. The paper gets researchers in the field to re-think about the pathogenesis in mucosa-related diseases, and it also raises a different approach at developing more effective prevention/treatment of mucosa-related diseases. I have few minor comments.

- 1) Authors can provide a schematic image, to better explain the complex interactions of nanoparticles in the hydrogel system.
- 2) How can the Matrigel better mimic the biological mucus, sputum or other hydrogels? What is the pore size and size distribution of the Matrigel network? Will they be similar to the biological hydrogels? Will the biotinylation level have any effects on the interaction?
- 3) Anchor-matrix bonds that are rapid and short-lived relative to anchor nanoparticulate bonds greatly enhance the trapping potency of molecular anchors. For most pathogens, their sizes are typically smaller than 200 nm (the diameter of PS-PEG used in this paper). Will the trapping trend of IgG applicable to much smaller nanoparticles (those can mimic virus in diameter)? What is the reason only testing 200 nm PS-PEG? Should authors test more different sized nanoparticles (size down to tens nanometers)?
- 4) What is the strength level of anti-PEG-IgG binding with PEG in comparison to the interaction of IgG binding with pathogens? How can the experimental data achieved based on anti-PEG-IgG binding with PS-PEG be translational to molecular anchors binding with pathogens?

Reviewer #2 (Remarks to the Author):

Contrary to the intuitively reasonable assumption that anchors with long-lived, high-affinity bonds to both the nanoparticulate and matrix would confer superior trapping efficiency, the authors showed experimentally and theoretically that anchor-matrix bonds that are rapid and short-lived relative to anchor-nanoparticulate bonds can greatly enhance the trapping efficiency of nanoparticles. Although the mathematical model does seem to explain the observed results, there are concerns regarding how widely applicable this artificial system based on anti-PEG-IgG and PS-PEG interactions and neutravidin-biotin is in physiological conditions. More extensive experimental conditions should be tested to support the stated conclusions.

- 1) The authors measured the binding of IgG and IgG-biotin to the gel treated with neutravidin, respectively. Compared with IgG, IgG-biotin showed higher binding affinity to the gel matrix treated with neutravidin. However, there is no measurement for the binding of IgG, IgG-biotin to the PEG-

nanoparticulate, especially in the presence of the gel matrix. As IgG-biotin may preferentially bind to the matrix treated with neutravidin, is it possible that IgG-biotin showed lower binding affinity to the nanoparticulate compared with IgG, especially in the presence of the gel matrix? This is important to confirm, as compromised IgG-biotin binding to the nanoparticulate may invalidate the experiments and the mathematic model as well.

2) The authors established a mathematical model, which is highly dependent on 6 assumptions, to recapitulate the observed results. Two assumptions are quite confusing and seem contradictory to the calculation. In assumption 2, the authors assumed that nanoparticulates have multiple (N) independent binding sites so that multiple anchors can simultaneously crosslink the same nanoparticulate to the matrix: $N \gg 1$. In assumption 6, the authors mentioned anchor concentration [A] is modest, such that on average a single nanoparticulate will not be simultaneously bound by two anchors that are immobilized to the matrix. Assumption 2 was later confirmed in the manuscript. However, for assumption 6, the rationale is not well supported. Isn't it obvious that at very low IgG concentration, matrix-bound IgG-biotin wouldn't have enough interaction with PEG-particles to trap them? A range of IgG concentrations should be tested to validate the results and extend the findings to physiological condition.

3) Neutravidin-biotin interaction is quite strong and no endogenous IgG would have such strong affinity toward matrix protein. Hence, this is quite an artificial system that should be studied with various IgGs with various affinities between anchor-matrix.

4) The authors used an artificial system which includes gel, anchors, and nanoparticulate (PS-PEG) as three crucial components to establish the mathematic model. Although this simplified mathematic model can explain the observed results by using 2 different anchors IgG and IgG-biotin, the authors didn't test whether this model is predictive for new anchors with different binding affinities to the matrix to validate their model. Importantly, it's unknown whether this model is valid under realistic conditions for other nanoparticulate species, most of which may have different properties such as surface property and intrinsic mobility in the gel compared with PS-PEG. In addition, various IgG-epitope affinities (including viruses) should be tested since extending the current findings on anti-PEG-IgG to other IgGs is tenuous.

Reviewer #3 (Remarks to the Author):

This is a very nice and high quality paper. The authors study a paradigmatic and biologically very important problem of a biogel matrix (e.g. mucus) populated with various nanoparticles (e.g. viruses) and anchor molecules (e.g. antibodies) which can bind to both the matrix and nanoparticles and make cross-links resulting in the nanoparticles trapping. The authors show that a very rapid binding/unbinding kinetics of anchor molecules to the matrix, i.e. making short-living weak bonds, can enhance the trapping potency of anchor molecules. This finding is contrary to some intuitive expectations that binding to both the nanoparticles and the matrix should be strong enough to ensure an efficient trapping. It provides a new paradigm. The authors did both experimental research and developed a pertinent stochastic theory of underlying diffusion-reaction processes. The theory nicely corroborates experimental findings. This is a high-quality important paper which I strongly recommend for publication in the Nature Communications.

Minor remark:

On p. 11, the authors state that they are doing a time-averaging of mean squared displacements

(MSD). The line formula displays, however, a squared displacement only, without any averaging in its lhs. I presume that the time-averaging over the variable t is simply missed by a mistake (typo). Please fix this typo. Another related question: Was also an additional ensemble averaging done, after the time averaging? The related results presented look rather smooth, which suggests that possibly yes. This point should be clarified in the amended manuscript.

Response to "A blueprint for robust crosslinking of mobile species in biogels using third-party molecular anchors with short-lived anchor-matrix bonds"

Reviewer #1 (Remarks to the Author):

Newby and others examined the characteristics of IgG (a representative third party molecular anchor) that could potentially maximize the net adhesive interactions between nanoparticulates and biopolymer hydrogel matrix. They found a mechanism, different from previously widely held assumption, that only rapid and short-lived anchor-nanoparticulate bonds (not the long-lived and high-affinity bonds) will greatly enhance the trapping potency of the molecular anchors. They applied a mathematic modeling to propose the unique mechanism that has been evaluated through experimental data. Their findings are novel and will be interesting to the scientific community specifically in mucus biology and mucosa-related pathophysiology. The findings form the foundation for better design and development of more potent anchors that can enhance the pathogen tracking in the biological gel system. The paper gets researchers in the field to re-think about the pathogenesis in mucosa-related diseases, and it also raises a different approach at developing more effective prevention/treatment of mucosa-related diseases. I have few minor comments.

Response: We thank the reviewer for their time and effort to review our work.

1) Authors can provide a schematic image, to better explain the complex interactions of nanoparticles in the hydrogel system.

Response: We agree a schematic image would greatly help convey the concept, and have now included a schematic as suggested (Fig. 1).

2) How can the Matrigel better mimic the biological mucus, sputum or other hydrogels? What is the pore size and size distribution of the Matrigel network? Will they be similar to the biological hydrogels? Will the biotinylation level have any effects on the interaction?

Response: Matrigel is actually a naturally occurring extracellular matrix (ECM) of the basement membrane that is similar to other biological hydrogels, which is why it is very widely used in biology. Its pore size has been reported by many investigators, generally ranging from as large as $\sim 2 \mu\text{m}$ at concentrations of $5\text{-}6 \mu\text{g}/\text{mL}^1$ to as small as $\sim 140 \text{ nm}$ at concentrations $\sim 12 \text{ mg}/\text{mL}^2$. In comparison, the average pore size in human cervicovaginal mucus is $\sim 300\text{-}400 \text{ nm}$, and ranges from $< 50 \text{ nm}$ to $> 2 \mu\text{m}^3$. Our addition of antibodies into Matrigel directly mimics the ECM environment in the body, since antibodies extravasate from the systemic circulation and are found throughout the interstitia where ECM are present. To ensure that biotinylation does not alter the microstructure or the diffusive barrier properties of Matrigel, we now included a comparison of the diffusion of both PEG-modified (i.e. inert) latex nanoparticles in biotinylated and native Matrigel preparations with and without anti-PEG IgG (Supp. Fig. 1). As the data shows, biotinylation does not alter the microstructure of the matrix, and we observed trapping potency comparable to native Matrigel preparations.

3) Anchor-matrix bonds that are rapid and short-lived relative to anchor nanoparticulate bonds greatly enhance the trapping potency of molecular anchors. For most pathogens,

their sizes are typically smaller than 200 nm (the diameter of PS-PEG used in this paper). Will the trapping trend of IgG applicable to much smaller nanoparticles (those can mimic virus in diameter)? What is the reason only testing 200 nm PS-PEG? Should authors test more different sized nanoparticles (size down to tens nanometers)?

Response: We thank the reviewers for pointing out this shortcoming. We had initially included ~200 nm PEG-coated nanoparticles because they were comparable in size to many mammalian viruses such as Herpes Simplex Virus (d ~ 180 nm), respiratory syncytial virus (d ~150-300 nm) and metapneumovirus (d ~ 200 nm). We have now included videos that showed smaller ~100 nm PEG-coated nanoparticles likewise diffuse freely in Matrigel, and that they are also effectively immobilized by addition of anti-PEG IgG (Supp. Videos 1-2). The ~100 nm PEG-coated nanoparticles are comparable in size to many viruses, including HIV (d ~ 90-120 nm) and influenza (d ~ 80-120 nm). We have also included videos showing that Respiratory Syncytial Virus (RSV) diffuses freely in Matrigel and is immobilized by the addition of Synagis, an anti-RSV IgG cocktail (Supp. Videos 3-4). We are unable to track smaller viruses with 2D particle tracking because they diffuse too quickly in and out of the focal plane to gain statistically robust inferences.

4) What is the strength level of anti-PEG-IgG binding with PEG in comparison to the interaction of IgG binding with pathogens? How can the experimental data achieved based on anti-PEG-IgG binding with PS-PEG be translational to molecular anchors binding with pathogens?

Response: The anti-PEG IgG used in this experiment has an affinity to PEG of $\sim 9 \times 10^{-9} \text{M}$ (i.e. 9nM), which is comparable to naturally occurring antibodies against many different pathogens. For example, common monoclonal antibodies that bind to gp140-trimer on HIV has comparable if not greater affinity than the anti-PEG IgG used here: 3.4 nM (VRC01), 1.3 nM (2G12), and 5.1 nM (PGT123), respectively⁴. Similarly, the three anti-Ebola antibodies in ZMapp, 2G4, 4G7, and 13C6, have apparent affinities to Ebola 0.9-7.0 nM⁵. To illustrate the relevance of our studies with anti-PEG IgG and PS-PEG beads, we have now included videos for respiratory syncytial virus (RSV) that showed viruses undergoing rapid diffusion in the same Matrigel setup treated with control IgG, but effectively immobilized when treated with RSV-specific IgG (Synagis®; Supp Videos 3-4).

Reviewer #2 (Remarks to the Author):

Contrary to the intuitively reasonable assumption that anchors with long-lived, high-affinity bonds to both the nanoparticulate and matrix would confer superior trapping efficiency, the authors showed experimentally and theoretically that anchor-matrix bonds that are rapid and short-lived relative to anchor-nanoparticulate bonds can greatly enhance the trapping efficiency of nanoparticles. Although the mathematical model does seem to explain the observed results, there are concerns regarding how widely applicable this artificial system based on anti-PEG-IgG and PS-PEG interactions and neutravidin-biotin is in physiological conditions. More extensive experimental conditions should be tested to support the stated conclusions.

Response: We thank the reviewer for their time and effort to review our work and their constructive criticisms. We have performed additional experiments wherever possible to address the identified shortcomings, and revised the manuscript accordingly.

1) The authors measured the binding of IgG and IgG-biotin to the gel treated with neutravidin, respectively. Compared with IgG, IgG-biotin showed higher binding affinity to the gel matrix treated with neutravidin. However, there is no measurement for the binding of IgG, IgG-biotin to the PEG-nanoparticulate, especially in the presence of the gel matrix. As IgG-biotin may preferentially bind to the matrix treated with neutravidin, is it possible that IgG-biotin showed lower binding affinity to the nanoparticulate compared with IgG, especially in the presence of the gel matrix? This is important to confirm, as compromised IgG-biotin binding to the nanoparticulate may invalidate the experiments and the mathematic model as well.

Response: We thank the reviewer for this important suggestion. The native anti-PEG IgG and biotinylated anti-PEG IgG used in our experiments *both* possess affinities of $\sim 9 \times 10^{-9}$ M (i.e. 9nM) to PEG (see Supp. Table 2). We are not aware of methods for binding kinetics/affinity measurements that could directly measure the binding affinity of antibodies to its antigen (PEG) embedded in a viscoelastic environment such as Matrigel. However, using biolayer interferometry, we did verify that the association of biotinylated anti-PEG IgG to neutravidin did not prevent it from binding to PEG (see Supp. Table 2). Our mathematical model predicts that matrix-bound anchors bind particles at a much lower rate than free unassociated anchors, not because the matrix-association directly precludes the anchor from binding the particle, but because matrix-bound anchors lose their ability to undergo rapid diffusion to collide and quickly accumulate on their corresponding antigen. In good agreement with this, with prolonged overnight incubation (Supp. Fig. 3), we observed a greater extent of nanoparticles become immobilized in biotinylated Matrigel with immobilized anchors (i.e. biotinylated anti-PEG IgG).

2) The authors established a mathematical model, which is highly dependent on 6 assumptions, to recapitulate the observed results. Two assumptions are quite confusing and seem contradictory to the calculation. In assumption 2, the authors assumed that nanoparticulates have multiple (N) independent binding sites so that multiple anchors can simultaneously crosslink the same nanoparticulate to the matrix: $N \gg 1$. In assumption 6, the authors mentioned anchor concentration [A] is modest, such that on average a single nanoparticulate will not be simultaneously bound by two anchors that are immobilized to the matrix. Assumption 2 was later confirmed in the manuscript. However, for assumption

6, the rationale is not well supported. Isn't it obvious that at very low IgG concentration, matrix-bound IgG-biotin wouldn't have enough interaction with PEG-particles to trap them? A range of IgG concentrations should be tested to validate the results and extend the findings to physiological condition.

Response: We thank the reviewer for pointing out this lack of clarity re Assumption #6. Our inclusion of Assumption #6 was to explicitly address the extreme (even if implausible) case where the proposed model would fail. Specifically, if there are such high concentrations of anchors that their average separations are smaller than the dimensions of a single nanoparticle, then the nanoparticles would most certainly be captured simultaneously by two or more anchors, regardless of their actual anchor-matrix affinities (i.e. even matrix-bound anchors would be very effective at trapping). However, in reality, such concentrations are simply never observed in biological systems. Indeed, in order for two immobilized anchors to be separated by a distance equal to the diameter of the nanoparticle (let's say $d \sim 100$ nm), the concentration of a single species of anchor (assuming it is an IgG antibody) must exceed $250 \mu\text{g/mL}$. In comparison, the concentration of most antigen-specific antibodies in mucus and in the body typically ranges from $1\text{-}20 \mu\text{g/mL}$.⁶ We believe it is prudent to include assumption #6 simply to delineate the boundaries of the model, but we are happy to remove it if the reviewer and editor feels that would be more appropriate.

Finally, we also want to clarify that both weak-affinity and matrix-bound IgG experiments are performed at an effective concentration ($10 \mu\text{g/mL}$) in Matrigel substantially above the IgG dissociation constant with PEG ($K_D \sim 1.35 \mu\text{g/mL}$). Furthermore, the concentration of IgG we used was by design already within the physiological range of naturally occurring antigen-specific antibodies, and hence our findings are entirely physiological. Since matrix-bound IgG already fails to trap at the highest concentrations tested, we did not include results with lower concentrations of matrix-bound IgG.

3) Neutravidin-biotin interaction is quite strong and no endogenous IgG would have such strong affinity toward matrix protein. Hence, this is quite an artificial system that should be studied with various IgGs with various affinities between anchor-matrix.

Response: We absolutely agree with the reviewer that no endogenous IgG would have the same affinity towards the matrix as neutravidin-biotin affinity. The experiment reported was done entirely to overturn the dogma that very strong anchor-matrix affinity would directly lead to greater trapping potency, and to guide future development of optimal anchors by establishing what is likely the optimal range of anchor-matrix affinities. Consistent with the hypothesis of rapid and weak anchor-matrix interactions and our predicted optimal anchor-matrix affinity, we have previously shown that anti-PEG IgM (an antibody slowed $\sim 30\text{-}50\%$ in mucus compared to in buffer i.e. $\eta \sim 0.5\text{-}0.7$) was able to trap PEG-nanoparticles in mucus more effectively and at lower concentrations than anti-PEG IgG (which is slowed $\sim 10\text{-}20\%$ in mucus compared to in buffer, i.e. $\eta \sim 0.8\text{-}0.9$)^{7,8}. We obtained similar results with Matrigel. However, because anti-PEG IgM also possesses greater binding avidity to PEG than anti-PEG IgG, we cannot directly conclude that the greater trapping potency is attributed entirely to differences in anchor-matrix affinity. We agree with the reviewer that it would further strengthen the concept of weak affinity matrix-anchor interactions if we can perform studies with IgGs with various anchor-matrix affinities. Unfortunately, there is currently no established experimental method to

precisely tune the affinity of IgGs to a matrix. Indeed, until this work, there was never a sound scientific rationale for such fine tuning of anchor-matrix interactions. We are currently actively investigating methods to engineer IgG with the predicted optimal anchor-matrix interactions, and we fully expect that, once published, our manuscript would motivate other investigators to do the same.

4) The authors used an artificial system which includes gel, anchors, and nanoparticulate (PS-PEG) as three crucial components to establish the mathematic model. Although this simplified mathematic model can explain the observed results by using 2 different anchors IgG and IgG-biotin, the authors didn't test whether this model is predictive for new anchors with different binding affinities to the matrix to validate their model. Importantly, it's unknown whether this model is valid under realistic conditions for other nanoparticulate species, most of which may have different properties such as surface property and intrinsic mobility in the gel compared with PS-PEG. In addition, various IgG-epitope affinities (including viruses) should be tested since extending the current findings on anti-PEG-IgG to other IgGs is tenuous.

Response: As discussed above in (3), there is currently no method that would allow us to study with new anchors with tunable anchor-matrix affinity while possessing the same anchor-antigen affinity. Our published observations anti-PEG IgM facilitates more effective trapping than anti-PEG IgG is consistent with the prediction by our model^{7,8}. The binding affinity of anti-PEG IgG to PEG ($K_D \sim 9\text{nM}$) is comparable to naturally occurring antibodies against many different pathogens. For example, common monoclonal antibodies that bind to gp140-trimer on HIV has comparable if not greater affinity than the anti-PEG IgG used here: 3.4 nM (VRC01), 1.3 nM (2G12), and 5.1 nM (PGT123), respectively⁴. Similarly, the three anti-Ebola antibodies in ZMapp, 2G4, 4G7, and 13C6, have apparent affinities to Ebola 0.9-7.0 nM⁵. The size of the nanoparticles studied ($\sim 100\text{ nm}$ and $\sim 200\text{ nm}$) are also comparable to the majority of mammalian viruses, and the concentrations of IgG used are comparable to virus-specific IgG that can be found in naturally occurring biological gels⁹. We have also shown that the anti-PEG IgG/PEG-beads system used here were able to trap PEG beads in mucus⁸ similar to virus-specific IgG immobilizing specific viruses^{9,10}. To validate our model by demonstrating that native IgG with weak-anchor affinity can mediate trapping of viruses in Matrigel, we have now included movies showing that RSV that otherwise rapid diffusion in Matrigel is effectively immobilized by the addition of RSV-specific IgG (Synagis; Supp. Videos 3-4). We believe these result, combined with earlier published findings showing IgG can trap viruses in different mucus secretions (lung, genital) and in different biological systems (human, mouse), strongly underscore that weak IgG-matrix interactions can facilitate effective trapping of nano-sized entities under a wide range of realistic and physiological conditions.

Reviewer #3 (Remarks to the Author):

This is a very nice and high quality paper. The authors study a paradigmatic and biologically very important problem of a biogel matrix (e.g. mucus) populated with various nanoparticles (e.g. viruses) and anchor molecules (e.g. antibodies) which can bind to both the matrix and nanoparticles and make cross-links resulting in the nanoparticles trapping. The authors show that a very rapid binding/unbinding kinetics of anchor molecules to the matrix, i.e. making short-living weak bonds, can enhance the trapping potency of anchor molecules. This finding is contrary to some intuitive expectations that binding to both the nanoparticles and the matrix should be strong enough to ensure an efficient trapping. It provides a new paradigm. The authors did both experimental research and developed a pertinent stochastic theory of underlying diffusion-reaction processes. The theory nicely corroborates experimental findings. This is a high-quality important paper which I strongly recommend for publication in the Nature Communications.

Minor remark:

On p. 11, the authors state that they are doing a time-averaging of mean squared displacements (MSD). The line formula displays, however, a squared displacement only, without any averaging in its lhs. I presume that the time-averaging over the variable t is simply missed by a mistake (typo). Please fix this typo.

Another related question: Was also an additional ensemble averaging done, after the time averaging? The related results presented look rather smooth, which suggests that possibly yes. This point should be clarified in the amended manuscript.

Thank you for pointing out this mistake. We have fixed the typo and added clarification about ensemble averaging. The text now reads:

“Under the assumption that particle paths are samples of a stationary stochastic process, mean squared displacement (MSD) can be computed by time averaging, i.e., $\langle \Delta r^2(\tau) \rangle = \sum_t \{ [x(t + \tau) - x(t)]^2 + [y(t + \tau) - y(t)]^2 \}$, (where τ = time scale or time lag). MSDs from all particle paths within a given video were then ensemble averaged to obtain $\langle \text{MSD} \rangle$ ”

References

- 1 Zaman, M. H. *et al.* Migration of tumor cells in 3D matrices is governed by matrix stiffness along with cell-matrix adhesion and proteolysis. *Proc Natl Acad Sci U S A* **103**, 10889-10894 (2006).
- 2 Tomasetti, L., Liebl, R., Wastl, D. S. & Breunig, M. Influence of PEGylation on nanoparticle mobility in different models of the extracellular matrix. *European Journal of Pharmaceutics and Biopharmaceutics* **108**, 145-155 (2016).
- 3 Lai, S. K., Wang, Y.-Y., Hida, K., Cone, R. & Hanes, J. Nanoparticles reveal that human cervicovaginal mucus is riddled with pores larger than viruses. *Proc Natl Acad Sci U S A* **107**, 598-603 (2010).
- 4 Yasmeen, A. *et al.* Differential binding of neutralizing and non-neutralizing antibodies to native-like soluble HIV-1 Env trimers, uncleaved Env proteins, and monomeric subunits. *Retrovirology* **11**, 41 (2014).
- 5 Davidson, E. *et al.* Mechanism of binding to Ebola virus glycoprotein by the ZMapp, ZMAb, and MB-003 cocktail antibodies. *J Virol* **89**, 10982-10992 (2015).
- 6 Wang, Y.-Y. *et al.* IgG in cervicovaginal mucus traps HSV and prevents vaginal herpes infections: Supplementary materials & methods. *Mucosal Immunol* **7** (2014).
- 7 Olmsted, S. S. *et al.* Diffusion of macromolecules and virus-like particles in human cervical mucus. *Biophys J* **81**, 1930-1937 (2001).
- 8 Henry, C. E. *et al.* Anti-PEG antibodies alter the mobility and biodistribution of densely PEGylated nanoparticles in mucus. *Acta Biomater* **43**, 61-70 (2016).
- 9 Wang, Y.-Y. *et al.* IgG in cervicovaginal mucus traps HSV and prevents vaginal herpes infections. *Mucosal Immunol* **7**, 1036-1044 (2014).
- 10 Wang, Y.-Y. *et al.* Influenza-binding antibodies immobilise influenza viruses in fresh human airway mucus. *Eur Respir J* **49**, 1601709 (2017).

REVIEWERS' COMMENTS:

Reviewer #1 (Remarks to the Author):

Thanks for the point-to-point responses and more experiments to address reviewer's comments. It is a very nice manuscript, and I would recommend for acceptance to publish.

Reviewer #2 (Remarks to the Author):

The authors have fully addressed this reviewer's comments and have provided nice, additional data that support their claims.

Reviewer #3 (Remarks to the Author):

I keep my opinion that this is a very nice and important paper which combines both theoretical and experimental research providing a new insight into the mechanisms of robust crosslinking of mobile species in biogels. The authors addressed my minor remark and amended their manuscript accordingly. It deserves publishing in the Nature Communications. I found, however, one misprint in the time averaging on p. 12, lines 3 and 4. Namely, the time window length, the normalization factor is absent. This obvious typo must be corrected before publication. I do not need to see the manuscript once again.

Response to " A blueprint for robust crosslinking of mobile species in biogels with weakly adhesive molecular anchors " (resubmission)

Reviewer #1 (Remarks to the Author):

Thanks for the point-to-point responses and more experiments to address reviewer's comments. It is a very nice manuscript, and I would recommend for acceptance to publish. No revisions are necessary. We thank the reviewer for their time and effort to review our work.

Reviewer #2 (Remarks to the Author):

The authors have fully addressed this reviewer's comments and have provided nice, additional data that support their claims. No revisions are necessary. We thank the reviewer for their time and effort to review our work.

Reviewer #3 (Remarks to the Author):

I keep my opinion that this is a very nice and important paper which combines both theoretical and experimental research providing a new insight into the mechanisms of robust crosslinking of mobile species in biogels. The authors addressed my minor remark and amended their manuscript accordingly. It deserves publishing in the Nature Communications. I found, however, one misprint in the time averaging on p. 12, lines 3 and 4. Namely, the time window length, the normalization factor is absent. This obvious typo must be corrected before publication. I do not need to see the manuscript once again.

We have addressed this typo. The equation now reads, " $\langle \Delta r^2(n\Delta t) \rangle = \frac{1}{N-n} \sum_{m=1}^{N-n} \{ [x((m+n)\Delta t) - x(m\Delta t)]^2 + [y((m+n)\Delta t) - y(m\Delta t)]^2 \}$." We thank the reviewer for their time and effort to review our work.